# Iatrogenic Injury of Biliary Tree—Single-Centre Experience

**DOI:** 10.3390/ijerph20010781

**Published:** 2022-12-31

**Authors:** Łukasz Nawacki, Monika Kozłowska-Geller, Monika Wawszczak-Kasza, Justyna Klusek, Przemysław Znamirowski, Stanisław Głuszek

**Affiliations:** Collegium Medicum, The Jan Kochanowski University in Kielce, 25-369 Kielce, Poland

**Keywords:** bile duct injury, cholecystolithiasis, cholecystectomy, complications, critical view of safety

## Abstract

Cholecystolithiasis is among the most prevalent gastrointestinal disorders requiring surgical intervention, and iatrogenic damage to the bile tree is a severe complication. We aimed to present the frequency of bile duct injuries and how our facility handles these complications. We retrospectively analyzed bile duct injuries in patients undergoing surgery. We concentrated on factors such as sex, age, indications for surgery, type of surgery, primary procedure, bile tree injury, repair, and timing as well as early and late complications. There were 22 cases of bile duct injury in the studied material, primarily affecting women—15 individuals (68.2%). Eleven cases (45.7%) of acute cholecystitis were the primary reason for surgery, and an injury to the common bile duct that extended up to 2 cm from the common hepatic duct was the most common complication (European Association for Endoscopic Surgery grade 2). Roux-en-Y hepaticojejunostomy was the most common repair procedure in 14 cases (63.6%). Eleven patients (50%) experienced early complications following reconstruction surgery, whereas five patients (22.7%) experienced late complications. An annual mortality rate of 22.7% (five patients) was observed. Iatrogenic bile duct injury is a severe complication of surgical treatment for cholecystolithiasis. Reconstruction procedures are characterized by high complication rates and high mortality.

## 1. Introduction

Cholecystolithiasis is a common gastrointestinal disorder affecting 10–15% of the adult population [1,2]. Symptomatic cholecystolithiasis is present in approximately 20% of patients and is a recurring indication of hospital admission for gastrointestinal disorders [3,4]. The gold standard of care for symptomatic cholecystolithiasis is laparoscopic cholecystectomy (LC). This type of surgery is one of Poland’s most frequently performed procedures. In 2020, 51,552 cholecystectomies (including 44,060 minimally invasive procedures) were performed, making up 136.02 interventions per 100,000 inhabitants [5]. Undisputed evidence supports the superiority of LC over open cholecystectomy (OC); however, as LC procedures increased, the rate of iatrogenic bile duct injuries (BDIs) rose from 0.2% to 1.5% [6]. Since LC was introduced, numerous attempts have been made to standardize this technique to reduce the complication rates [7].

The Critical View of Safety (CVS), which was first described in 1995, is considered to be the standard method for identification of the cystic structures [8]. Briefly, we should visualize two tubular structures connected to the gallbladder—the cystic duct and the cystic artery. Standardization of LC resulted in a reduction of the BDI rate from 0.3% to 0.23% and even 0.09% if the CVS was properly adopted [8,9,10]. Nevertheless, the introduction of CVS has not been universal, and, what’s more, some surgeons don’t properly understand how to achieve the critical view of safety [11]. Therefore, the injury to the biliary tree has not been excluded as a complication of cholecystectomy.

Iatrogenic injuries to the biliary tree are serious side effects of cholecystectomy that affect both immediate and long-term outcomes as well as the patient’s quality of life [12]. Several systems classify BDIs [13,14,15,16,17,18]. However, irrespective of which classification system is used, most authors agree that it is essential to intraoperatively recognize a BDI followed by immediate reconstruction, as this approach produces the best results [12,19]. Most patients may experience symptoms like bile leakage or jaundice, cholangitis, or sepsis when an injury is not identified during cholecystectomy. In such cases, some surgeons insist that the reconstruction procedure should be delayed until inflammation has resolved [20,21].

There are several possible reconstruction procedures for biliary tree injuries, the most common being endoscopic retrograde cholangiopancreatography (ERCP), primary end-to-end anastomosis (with or without Kehr’s T-tube drainage), choledochoduodenostomy, and choledochojejunostomy. Most centers prefer Roux-en-Y choledochojejunostomy to treat severe BDIs [22,23,24,25].

We sought to present the frequency of BDIs and techniques employed in our facility to handle such complications in response to the information provided above.

## 2. Materials and Methods

The data of patients who sustained iatrogenic BDI were collected on an ongoing basis [26]. The observation period spanned from 2000 to 2020. The type of surgery (emergency or elective), primary procedure and indications, BDI type, timing, length of hospital stay following reconstruction, early complications (up to 30 days following reconstruction), and follow-up after one year were assessed.

The collected data were statistically analyzed using R software version 4.1.1. The alpha level assumed in this study was α = 0.05. We examined the entire for qualitative variables, study group; n and % were used, whereas for qualitative variables, the mean with standard deviation or median with the 1st and 3rd quartiles were used. The normality of distribution was evaluated using the Shapiro–Wilk test. Relationships between two categorical variables were assessed using Fisher’s exact test. To compare variations in continuous variables between two groups, the Mann–Whitney U test was used.

## 3. Results

All the laparoscopic cholecystectomies were performed according to the critical view of safety. During the observation period there were 5341 cholecystectomies performed: 4493 LC and 848 OC (out of which 764 were conversions from LC). Acute cholecystitis was an indication for emergency surgery in 1525 cases from the LC group (33.9%) and 672 patients (79.2%) from the OC group. In the observation period there were 24 cases of bile duct injury, however in two cases the BDI occurred during a surgery unrelated to gallbladder. These patients were excluded from this research. Furthermore, patients with a BDI transferred from a different hospital were excluded from the study (4 cases). The inclusion criteria were any injury to the biliary tract during cholecystectomy performed in a tertiary hospital, independently of the time when it was discovered. Therefore, the study group consisted of 22 patients (0.4% of all cholecystectomies). The general characteristics of the study group are given in Table 1.

In most patients with iatrogenic BDI, the surgery was urgent—12 patients (54.5%); the main indication in this group was acute cholecystitis—11 patients (91.7%). The primary procedure in most patients (13 cases; 59%) was LC; however, LC accounted for 46.2% in emergency cases.

Table 2 shows the type of BDIs that occurred in our patients and their classification.

The most common site of BDI was the common hepatic duct—10 patients (45.4%), followed by the common bile duct—7 cases (31.8%). Ten patients had type 2 injuries, or injuries that were 2 cm or closer from the bile duct junction, according to the European Association for Endoscopic Surgery (EAES) classification.

Table 3 explains the circumstances of BDI during laparoscopic cholecystectomy.

Overall, the number of BDI during LC was 0.29%. As mentioned above, all the procedures were performed according to the CVS. The reason for an injury was technical error in four cases (30.77%) and misinterpretation of the cystic duct in 9 cases (69.23%)

In most patients, the BDI was recognized during the primary procedure (11 out of 22 patients; 50%), and in 19 cases (86.4%), the reconstruction procedure was performed within 2 weeks after the primary procedure. The reconstruction was performed in the remaining three cases days 17, 25, and 43 following the initial procedure (Table 4).

The most common reconstruction procedure was Roux-en-Y hepaticojejunostomy (14 patients), and closure with T-tube drainage was performed in 6 cases. The average hospital stay following reconstruction lasted 13.2 days.

Out of the 11 patients who underwent bile duct reconstruction during the primary procedure, eight had emergency indications, and more than half (6; 54.5%) underwent bile duct closure with T-tube drainage. Other patients underwent hepaticojejunostomy. Early complications did not occur in fewer than half of the patients (5; 45.5%), whereas early postoperative death occurred in three cases (days 2, 7, and 13 after the reconstruction procedure). The complications are summarized in Table 5. The most common surgery performed in patients with early complications was Roux-en-Y choledochojejunostomy (91%). With no early complications, this procedure was performed in 36% of patients, and the relationship between the variables was statistically significant (*p* = 0.043). No difference in time to reoperation was found between the no early complications and early complications groups (*p* = 0.900).

A total of five deaths occurred in the study group, all in the early postoperative period. In the early complications group, the cause of death was gastrointestinal bleeding (hemobilia) in two cases, septic shock in two cases, and diffuse peritonitis due to bile leakage in one case.

The most common type of injury, according to the EAES classification in patients with no early complications, was type 1 (36.4%), whereas in patients with early complications, it was type 2 (63.6%). The correlation, however, was not significant statistically (*p* = 0.335). Analyzing patients with or without complications after a year revealed no significant correlations or differences (*p* > 0.050 for all analyses). According to the EAES classification in patients with and without complications after one year, type 2 injury was most prevalent (60% and 42%, respectively).

Time to reoperation did not differ between both groups. In addition, there was no connection between complications after a year and the type of reconstruction procedure. Most patients from both groups underwent Roux-en-Y choledochojejunostomy (50% of patients without complications after one year and 80% of patients with complications after one year).

Death was also unrelated to any studied characteristics (*p* > 0.050 for all variables). Most surviving patients had a type 2 injury (47%), according to the EAES classification. According to the EAES classification, type 1 injuries represented 60% of patients who did not survive. The most common procedure performed in patients of both groups was Roux-en-Y choledochojejunostomy (58.9% of survivors and 80% of deaths). The time to reoperation did not differ between the groups studied.

## 4. Discussion

Bile duct injury is a severe complication of cholecystectomy, but fortunately, it is relatively rare. However, it requires reoperations, which increases mortality and reduces the quality of life.

In our study, most BDIs occurred in women (68.2%), and the mean age was 61.5 years. Our data is consistent with findings in other publications [10,27,28]. The primary cause of cholecystectomy in the study group was acute cholecystitis (50%), and urgent cholecystectomy in all cases was performed. In the available literature, complicated cholecystolithiasis was also an indication for cholecystectomy, including acute and chronic cystitis; however, only 20% were emergency surgeries [10]. Acute cholecystitis has been shown to increase the risk of BDIs [27].

More than half of the injuries occurred during laparoscopy (59.1%). The data in this study differ slightly from some of the available publications, where the majority of injuries happened during open surgery [28,29,30] because LC was already widespread during the observation period. Today, most cholecystectomies are performed laparoscopically, so BDIs typically occur during this type of intervention [24,28].

Most injuries affect the main bile ducts—the common hepatic and common bile ducts (45.5% and 31.8%, respectively), which are type 2 and type 1 injury, respectively, in the EAES classification [16]. In many publications, the Strasberg classification is used [29,31,32,33,34,35]. Both classifications are based on the location of the BDI; however, the EAES classification also considers several other parameters. This is the only system that evaluates seven factors, including the location and extent of the lesion, presence of additional vascular damage, partial excision of bile ducts, time of lesion identification, etiology of the lesion, and type of injury. To compare with other commonly used classifications, we primarily focused on evaluating the location of the BDI. The exact details of the full classification are given in Appendix A. As shown, the bile ducts were completely damaged in 13 cases, severely or slightly damaged in four cases, whereas in two cases, there was an injury to the accessory bile duct (duct of Luschka) or leakage of the bile duct stump. In our opinion, the EAES classification includes all the most significant factors considered in BDIs.

As previously mentioned and demonstrated in Table 1, in 11 patients (three elective and eight emergency cases) the bile duct injury was discovered during the primary procedure (in six cases the injury occurred during LC). In each case, the reconstruction procedure was performed as an open surgery during the same operation. In five cases, it was a Roux-en-Y reconstruction, whereas in six cases, it was primary bile duct closure with Kehr’s T-tube drainage. It has not been demonstrated in the currently available literature if the timing of the bile duct reconstruction affects postoperative complications [36]. There is a consensus that identifying the BDI during the procedure with immediate reconstruction of the bile duct brings the best results [10,12]. However, early reconstruction in a critically ill patient who is hemodynamically unstable or has sepsis is associated with a worse prognosis [12]. In contrast, in reference centers, there is no difference in outcomes between early and delayed reconstruction [33,37]. Roux-en-Y hepaticojejunostomy is preferred in most cases [38]. In our study, this was also the most frequently performed procedure in 14 out of 22 cases (63.6%).

In 11 patients, there were no complications within 30 days after the reconstruction (in 6 patients, an immediate reconstruction was performed; in 4 cases, the procedure was performed within 2 weeks; and in 1 patient, the procedure was postponed by 2 months). In five patients who developed complications, the BDI was identified during the primary procedure, and immediate reconstruction was performed. In four cases, the reconstruction was completed within two weeks, whereas it took more than 2 weeks for the other two patients. In the available literature, the most common complications following bile duct reconstruction include recurrent cholangitis (11.4–23%), anastomotic leakage (5–10%), intraabdominal abscess (14%), and anastomotic stricture (30%) [25,39]. In our study, the most frequent early complication was anastomotic leakage (22.7%), but none of the patients required reoperation. However, the most common complication during the first year was recurrent cholangitis (18.2%), which is consistent with other publications. Complications are linked to higher treatment costs and lower patient quality of life [28,31,33].

Many publications focused on preventing iatrogenic BDIs [6,16,40] and emphasized the necessity of accurate visualization of the cystic duct and artery (so-called critical view of safety) and proper traction of the fundus of the gallbladder. In our study, 50% of the iatrogenic BDIs occurred during cholecystitis surgery. In such cases, good visualization and traction may be difficult to achieve.

The limit of the study is the small sample size, however the number of BDI (0.4% overall, 0.29% in laparoscopic cholecystectomy) is similar to other studies.

## 5. Conclusions

Iatrogenic BDIs are among the most serious complications of cholecystectomy. We believe that a unified classification will help conduct further studies, which in turn will help reduce the rate of BDIs, and if such a complication occurs, we will be able to determine the best reconstruction technique and when it should be performed. This is essential because iatrogenic BDIs are linked to several sequelae, higher mortality, and lower quality of life.

## Figures and Tables

**Table 1 ijerph-20-00781-t001:** General characteristics of the study group.

Characteristics	Value
Sex (female), n (%)	15 (68.2)
Male, n (%)	7 (31.8)
Age (y), M ± SD	61.5 ± 15.06
**Indication for surgery, n (%)**	
Cholecystolithiasis	10 (45.4)
Acute cholecystitis	11 (50)
Polyp of the gallbladder	1 (4.5)
**Type of surgery, n (%)**	
Emergency	12 (54.5)
Elective	10 (45.4)
**Primary procedure, n (%)**	
Open cholecystectomy	9 (41)
Laparoscopic cholecystectomy	13 (59)

**Table 2 ijerph-20-00781-t002:** Classification of injury.

Site of Injury, n (%)	
Common hepatic duct	10 (45.4)
Right hepatic duct	3 (13.6)
Duct of Luschka	1 (4.5)
Common bile duct	7 (31.8)
Cystic duct	1 (4.5)
**Type of injury—EAES classification, n (%)**	
1	7 (31.8)
2	10 (45.4)
5	3 (13.6)
6	2 (9)

EAES, European Association for Endoscopic Surgery.

**Table 3 ijerph-20-00781-t003:** Cause of bile duct injury during the laparoscopic cholecystectomy.

Sex	Age (Years)	Indication for Surgery	Type of Surgery	Site of Injury	Cause of Injury
Female	29	Cholelithiasis	Elective	Common hepatic duct	Misinterpretation
Male	65	Cholelithiasis	Elective	Common bile duct	Misinterpretation
Female	57	Cholelithiasis	Elective	Right hepatic duct	Misinterpretation
Female	68	Cholelithiasis	Elective	The Duct of Luschka	Technical error
Female	34	Cholelithiasis	Elective	Common hepatic duct	Misinterpretation
Male	75	Cholelithiasis	Elective	Common hepatic duct	Misinterpretation
Female	84	Cholelithiasis	Elective	Common hepatic duct	Misinterpretation
Female	56	Cholelithiasis	Emergency	Common hepatic duct	Misinterpretation
Female	63	Acute cholecystitis	Emergency	Common hepatic duct	Misinterpretation
Female	41	Acute cholecystitis	Emergency	Common bile duct	Technical error
Male	35	Acute cholecystitis	Emergency	Common bile duct	Technical error
Male	64	Acute cholecystitis	Emergency	Cystic duct	Technical error
Female	82	Acute cholecystitis	Emergency	Common bile duct	Misinterpretation

**Table 4 ijerph-20-00781-t004:** Reconstructive surgery details.

Type of Reconstructive Surgery, n (%)	
ERCP—slight bile leakage near one of the clips; laparotomy and suture of the duct of Luschka	1 (4.5)
ERCP—plastic stent	1 (4.5)
Bile duct closure with T-tube drainage	6 (27.3)
Roux-en-Y hepaticojejunostomy	14 (63.6)
Time to reoperation (days), Me (Q1; Q3)	0.00 (0.00; 9.25)
Duration of hospital stay after reconstruction (days), Me (Q1; Q3)	10.0 (7.50; 14)

**Table 5 ijerph-20-00781-t005:** Complications of reconstructive surgery.

Early Complications (within 30 Days), n (%)	
None	11 (50)
Bile leakage	5 (22.7)
Cholangitis	1 (4.5)
Death	5 (22.7)
**Complications after one year, n (%)**	
None	12 (54.5)
Recurrent cholangitis	4 (18.2)
Anastomotic stricture, external drainage of the bile ducts	1 (4.5)

## Data Availability

Data available within the article or its Appendix A.

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
