# Peer review of "Iatrogenic Injury of Biliary Tree—Single-Centre Experience"

_ijerph, 2022, doi:10.3390/ijerph20010781_

Round 1
Reviewer 1 Report
Congratulations for your work!
I recommend you to detail the criteria for inclusion in the study.
In the 20 years of observation, how many laparoscopic cholecystectomies were performed and what was the percentage of complications?
What were the exclusion criteria?
What are the limits of the study?
Please, add all this answers în your article
Please, also add a list of abbreviations.
Author Response
Dear Reviewer,
We would like to thank you for taking the necessary time and effort to review the manuscript. We sincerely appreciate all your valuable comments and suggestions, which helped us in improving the quality of the manuscript.
In the 20 years of observation, how many laparoscopic cholecystectomies were performed and what was the percentage of complications?
Response: The answer has been uploaded in the manuscript – lines 83 – 86. During the observation period there were 5341 cholecystectomies performed: 4493 LC and 848 OC (out of which 764 were conversions from LC). Acute cholecystitis was an indication for emergency surgery in 1525 cases from the LC group (33.9%) and 672 patients (79.2%) from the OC group.
What were the exclusion criteria?
Response: The exclusion criterion was an injury to the biliary tract unrelated to a gallbladder surgery and BDI that was transferred to our hospital from a different facility. The answer has been given in the manuscript – lines: 86 – 89.
What are the limits of the study?
Response: The limit of the study is the small sample size, however the number of BDI (0.4% overall, 0.29% in laparoscopic cholecystectomy) is similar to other researches. Lines 219 - 220
Please, add all this answers în your article
Response: all the answers have been included in the test
Please, also add a list of abbreviations.
Response: a list of abbreviations has been uploaded at the beginning of the text
Reviewer 2 Report
This is a hot topic and I think always will be since laparoscopic cholecystectomy will remain one of the most frequent procedures in general surgery.
The introduction needs more recent data regarding the cognitive mechanisms involved in distinguishing the anatomy.
I would improve the study design by excluding the 2 cases not related to LC or OC as the conclusions revolve around cholecystectomy as the index intervention. More over, since the number of cases is small I would explain in a table the circumstances in which the injury occurred, this shouldn't be so difficult.
Should be difficult to explain how the injury occurred in 9 cases of OC, since there are a number of bailout procedures to resort to in difficult situations.
Author Response
Dear Reviewer,
We would like to thank you for taking the necessary time and effort to review the manuscript. We sincerely appreciate all your valuable comments and suggestions, which helped us in improving the quality of the manuscript.
The introduction needs more recent data regarding the cognitive mechanisms involved in distinguishing the anatomy.
Response: The introduction has been improved with the section regarding the Critical View of Safety – lines 44 – 51.
I would improve the study design by excluding the 2 cases not related to LC or OC as the conclusions revolve around cholecystectomy as the index intervention. More over, since the number of cases is small I would explain in a table the circumstances in which the injury occurred, this shouldn't be so difficult.
Response: The cases unrelated to gallbladder surgery have been excluded and an additional table has been included (table 3) that describes the reasons for BDI during laparoscopic cholecystectomy.
Should be difficult to explain how the injury occurred in 9 cases of OC, since there are a number of bailout procedures to resort to in difficult situations.
Response: Unfortunately, I was not able to describe the reasons for BDI during open cholecystectomy.